# ERM++: An Improved Baseline for Domain Generalization

## Abstract

Multi-source Domain Generalization (DG) measures a classifier's ability to generalize to new distributions of data it was not trained on, given several training domains. While several multi-source DG methods have been proposed, they incur additional complexity during training by using domain labels. Recent work has shown that a well-tuned Empirical Risk Minimization (ERM) training procedure, that is simply minimizing the empirical risk on the source domains, can outperform most existing DG methods. ERM has achieved such strong results while only tuning hyper-parameters such as learning rate, weight decay, and batch size. This paper aims to understand how we can push ERM as a baseline for DG further, thereby providing a stronger baseline for which to benchmark new methods. We call the resulting improved baseline **ERM++**, and it consists of better utilization of training data, model parameter selection, and weight-space regularization. ERM++ significantly improves the performance of DG on five multi-source datasets by over 5% compared to standard ERM using ResNet-50, and beats state-of-the-art despite being less computationally expensive. We also demonstrate the efficacy of ERM++ on the WILDS-FMOW dataset, a challenging DG benchmark. Finally, we show that with a CLIP-pretrained ViT-B/16, ERM++ outperforms ERM by over 10%, allowing one to take advantage of the stronger pre-training effectively. We will release code upon acceptance.

## 1 Introduction

Domain Generalization (DG) (Blanchard et al., 2011; Muandet et al., 2013) tackles the crucial task of developing models that can excel on previously unseen data distributions, all without relying on the availability of target data for model updates. This is vital when gathering new domain-specific data is impractical, and differences between training and deployment data are unknown beforehand. In *multi-source* DG, each training sample is categorized as belonging to one of several domains. Many advanced methods explicitly harness this domain membership information (Ganin et al., 2016; Zhang et al., 2021; Li et al., 2018b; Zhou et al., 2021). However, recently DomainBed (Gulrajani & Lopez-Paz, 2020) conducted a comprehensive evaluation of these methods and revealed that Empirical Risk Minimization (ERM) surprisingly outperforms most prior work in DG when hyperparameters are well-tuned. This achievement is particularly remarkable because ERM relies on domain labels in a rather limited manner, through oversampling minority domains to balance domain sizes in the training data. Even though advanced techniques come equipped with strong inductive biases, they fail to surpass ERM's performance. This shows the critical importance of well-tuned baselines; they ensure that research results are reliable and meaningful. Without a solid baseline, it can be challenging to determine whether reported improvements are due to the proposed method's effectiveness or simply a result of arbitrary choices or overfitting to the dataset. Nevertheless, Gulrajani & Lopez-Paz (2020)'s tuning of ERM only consists of learning rate, weight decay, and batch size. There are substantial other important design choices, and our primary objective in this paper is to examine those which do not result in alterations to model architecture or intricate training strategies.

We conduct a critical analysis of various components within the training pipeline to develop ERM++, revolving around three main themes. First, we explore how the training data is being used, including training length and checkpoint selection (Section 3.1). Currently, the DG standard is to split off a subset of data for hyper-parameter and checkpoint selection (Gulrajani & Lopez-Paz, 2020; Cha et al., 2021; 2022). Motivated by the metric learning literature (Mishra et al., 2021; Movshovitz-

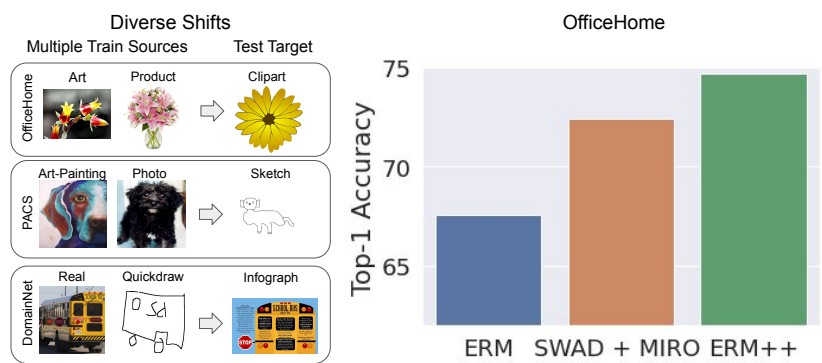

Figure 1: We tackle the task of Multi-Source DG, where a model is trained on several source domains and evaluated on a different target domain. This simulates the real-world scenario where we don't know how deployment data will differ from training data. We do this by improving the classic, and already strong, ERM (Gulrajani & Lopez-Paz, 2020) algorithm with careful application of known methodologies and call it ERM++. We verify our method on a diverse set of domain shifts, and show that ERM++ achieves the best reported numbers in the literature, and even outperforms the state-of-the-art SWAD + MIRO(Cha et al., 2021). We argue that ERM++ should become the default baseline to build off of.

Attias et al., 2017; Tan et al., 2019), we instead propose to two passes: one to select hyper-parameters and training length, and the second to retrain with the full-data and selected hyper-parameters. This allows us the leverage the fact that increasing data size improves generalization. Second, we consider how we initialize network parameters such as the selection of pretraining network and whether or not to fine-tune or freeze layers (Section 3.2). While pre-training has been shown to be critical for domain adaptation and generalization(Angarano et al., 2022; Kim et al., 2022), stronger initializaton has not yet become part of a standard DG benchmark. Similarly, while first training the linear probe has been widely used over the years (Kanavati & Tsuneki, 2021; Zhai & Wu, 2018), it has not been integrated into a commonly used multi-source DG baseline. Third, we investigate weight-space regularization methods that are often used to help avoid overfitting to the training data (Section 3.3). It's been shown that averaging model iterates results in converging to flatter minima and improved generalization(Arpit et al., 2021; Cha et al., 2021; Izmailov et al., 2018; Wortsman et al., 2022b), and we investigate different averaging strategies.

Put together, we find that we can outperform all prior work by over 1 percent on ResNet-50, when we hold 1 domain out in 5 datasets (OfficeHome, PACS, DomainNet, TerraIncognita, and VLCS), and average across domains. In the process, we show the robustness of two recent works (MIRO and DIWA) which do not use domain label information on top of ERM++, while also showing that a method which does use domain-labels (CORAL (Sun & Saenko, 2016)) doesn't compose well with ERM++. Finally, we demonstrate massive utility of ERM++ on CLIP-pretrained ViT(Radford et al., 2021) relative to ERM, giving an over 10% performance improvement on the same 5 tasks.

## 2 RELATED WORK

**Domain-invariant feature learning:** In multi-source domain generalization, it is common to leverage the domain labels to learn domain-invariant features. CORAL (Sun & Saenko, 2016) aligns second-order statistics of different domains. DANN (Ganin et al., 2016) uses an adversarial loss to match feature distributions across source domains. However, using domain knowledge to learn domain-invariant features can learn to ignore important signals. In fact, Vedantam et al. (2021) find low correlation between low source-target discrepancy and good DG performance.

**Domain-Aware Data Augmentation:** Data augmentation is a common tool to expand the training domain (Zhou et al., 2021; Hendrycks et al., 2020; Zhong et al., 2022; Yan et al., 2020). For example, Inter-domain mixup (Yan et al., 2020) blends the images of different domains, and augmentation with style transfer can further diversify training images (Zhong et al., 2022), though it is expensive. Instead of relying on data augmentation techniques during training on sources, we propose to employ all training samples from the source, including validation data, which expands knowledge about

the task. We also propose to use backbones pretrained with strong domain-agnostic augmentation such as Augmix (Hendrycks et al., 2020), which mixes different synthetic augmentations.

**Ensembling:** Deep ensembles are effective for domain generalization (Arpit et al., 2021; Fort et al., 2019). However, they are computationally inefficient, needing to run inference through many models. It has been recently shown that averaging model weights can approximate an ensemble(Wortsman et al., 2022a; Rame et al., 2022; Wortsman et al., 2022b; Cha et al., 2021; Izmailov et al., 2018), either from a single trajectory or multiple trajectories. We choose to leverage the ensembles from a single training trajectory.

**Preventing Catastrophic Forgetting:** Several recent approaches aim to leverage generalizable features from a model pre-trained on large-scale data. Adapting such a model to the downstream task without forgetting its generalizable representations is the key to achieving generalization Wortsman et al. (2022b) interpolate between the pre-trained and adapted model. Kumar et al. (2022) and Zhai & Wu (2018) mitigate feature distortion by pre-training a linear probe first before fine-tuning the backbone, warmstarting the fine-tuning with a good initialization. MIRO (Cha et al., 2022) maximizes the mutual information in feature space between the fine-tuned and pre-trained networks. Our approach utilizes warmstart and confirms its effectiveness in diverse settings.

## 3   REVISITING TRAINING PROCEDURES TO CREATE ERM++ FOR DOMAIN GENERALIZATION

We study the problem of Multi-Source DG for classification. We train a model on data consisting of multiple domains and evaluate it on data from unseen domains. More formally, let us consider training domains $d \in \{d_1, ..., d_n\}$. A training dataset is constructed using all sample, label pairs in all training domains $D = \{(X^{d_1}, Y^{d_1})...(X^{d_n}, Y^{d_n})\}$. After training classifier $f$ on $D$, it is tested on a held-out testing domain $d_{test}$. As stated in previous sections, approaches utilizing invariance of the domain or regularization of features can complicate the training. Instead we perform simple empirical risk minimization (ERM), formalized as minimizing the average loss over all samples $\frac{1}{n}\sum_{i \in D} \ell(x_i, y_i)$. In practice, we compose batches to be equal parts of each source domain.

Our goal is to investigate the general training components that go into creating an ERM model to provide a strong baseline for future work, ensuring that improvements reported by new methodolgies cannot be achieved using simpler means. These components include how to effectively use the source data (Section 3.1), considerations when selecting and using pretrained weights (Section 3.2), and weight-space regularization methods that help prevent overfitting to the source domains (Section 3.3). We refer to our new stronger baseline as ERM++.

### 3.1   IMPROVED DATA UTILIZATION

A key component of training any neural network is utilizing the (often limited) training data effectively. A common practice in the domain generalization literature is to split source datasets into (often 80%/20%) train/validation sets under a fixed number of iterations for each dataset (e.g. , (Gulrajani & Lopez-Paz, 2020; Cha et al., 2021; Rame et al., 2022; Arpit et al., 2021)). The validation data is used to set hyperparameters and perform checkpoint (no. training steps) selection. This approach has two major drawbacks. First, by creating a separate validation set we are sacrificing a significant portion of our labeled data, and data quantity is known to be important for generalization. Second, by training under a fixed (relatively small) number of iterations we ignore the varying convergence rates of different models, which may result in an underperforming model. We address these with techniques we call ***Long Training, Early Stopping*** and ***Full Data***.

**Allowing model convergence *(Long Training, LT)*:** We observe that source validation performance does not saturate on many datasets (See Appendix B.3), therefore we increase the celing on number of training steps by 4x. This allows the model to achieve its maximum performance.

**Determining training length *( Early Stopping, ES)*:** Given the high ceiling on number of training steps, it is possible to overfit. Therefore, we use validation performance to select the number of training steps. This number of training points is a parameter we call $\phi$. This is similar to checkpoint selection as done in prior work, however we afterwards retrain with the full data (see below).

**Using the full data *(Full Data, FD)*:** Inspired by the training procedures in metric learning literature (e.g. Mishra et al. (2021); Movshovitz-Attias et al. (2017); Tan et al. (2019); Teh et al. (2020); Wang et al. (2020)), we explore a two-stage training procedure. In the first stage we use the same

train/validation splits as in prior work in order to choose training length. In the second stage we train our model for deployment using the entire (train+validation) dataset in order to train with the entire dataset for better generalization.

## 3.2 Initializing Model Weights

Most domain generalization methods do not train a model from scratch, but rather transfer the weights of an existing model, typically pretrained on ImageNet (Deng et al., 2009). This is motivated by the idea that pre-training allows the model to learn robust features useful for generalization on downstream tasks. There are three main decisions that we explore further: selecting what model weights to transfer, determining what weights to fine-tune or keep frozen, and how to initialize any new weights (e.g. final classifier weights) in your network.

**Weight Initialization (*Strong Init.*):** Recent work has shown that better ImageNet models have better domain generalization properties for DG (Kim et al., 2022; Angarano et al., 2022). However, this has been explored in the context of varying model size. Therefore, performance gains can be either from a.) improved pre-training dataset (upstream) performance resulting in improved DG or b.) larger models resulting in improved DG performance, regardless of upstream performance. These also disregard the needs of some applications, such as computational requirements (larger models necessitate more resources) or restrictions on architectures due to a shared encoder for a multitask problem. Thus, we explore the effect of different initializations for the same model architecture, specifically a ResNet-50 (He et al., 2016):

- **TorchVision Model Weights:** This is the standard ImageNet pretrained initialization present in TorchVision. It was trained with weak augmentations for 90 epochs.
- **AugMix trained network**: AugMix (Hendrycks et al., 2020) is a method used to improve model consistency using augmentations without training the model on data which is too different from the test data. AugMix takes two augmented views of an image and mixes them in pixel space. Then the model is trained to produce consistent output between two AugMix augmentations and the clean image.
- **ResNet A1:** ResNet A1 initializes weights from the training recipe presented in (Wightman et al., 2021). The model is heavily tuned to find training settings which result in very strong ImageNet performance. Examples include training for 600 epochs, the LAMB optimizer, strong augmentations, and a binary cross-entropy.
- **Meal V2** : MealV2 (Shen & Savvides, 2020) is a highly performant ensemble, distilled into a ResNet-50. In particular, a SeNet-154 (Hu et al., 2018) (81.23% ImageNet Top-1) and ResNet-152 (81.02% ImageNet Top-1) are distilled into ResNet-50.

Each of these models has different ImageNet validation accuracies, ranging from 76.13% (TorchVision weights) to 80.7% (Meal-V2 (Shen & Savvides, 2020)). However, as our experiments will show, simply swapping out the standard initialization for the strongest ImageNet model does not result in the best performance. We empirically find the strongest of these, Augmix (Hendrycks et al., 2020), and refer to it as *Strong init*.

**Unfreezing BatchNorm(*UBN*):** It has been shown that what parameters to update during fine-tuning a pre-trained model, and when, can have substantial effects on downstream performance. Surgical fine-tuning (Lee et al., 2022) shows that only updating some blocks results in improved performance, but that different datasets require the unfreezing of different blocks, making it unsuitable for a general DG training procedure (as is our goal). Therefore, most domain generalization methods will fine-tune most layer weights, with the exception of BatchNorm parameters, which are sometimes kept frozen. We experiment further with the effect freezing the BatchNorm parameters has on performance, and refer to unfreezing them as *UBN*.

**Initializing Classifier Weights (*Warm Start, WS*):** New class labels require new a classification layer, and a standard method is to initialize a new classifier randomly and subsequently finetune the entire model. However, a recurring observation made by many researchers over the years is that your model may suffer from divergence from the initialization due to the noisy gradients from newly initialized layers (Goyal et al., 2017; He et al., 2016; Rame et al., 2022). In the case of pretrained models, this results in catastrophic forgetting of robust, pre-trained features. To address this, researchers would begin training by Warmstart (*WS*) (Kanavati & Tsuneki, 2021; Zhai & Wu, 2018) (also commonly referred to as warmup), where the new layer weights are trained with all pretrained weights kept frozen for several hundred steps. After this short training cycle, new and old layer weights are finetuned together (sometimes except for BatchNorm layers).

|  | Hparam Search Runs | Train FLOPS | Avg. Top-1 |
|---|---|---|---|
| ERM++ w/out *LT*(ours) | 2 (for *ES*) | 1x | 68.4% |
| ERM++(ours) | 2 (for *ES*) | <4x | **68.9**% |
| MIRO | 4 (for $\lambda$) | 2x | 68.1% |
| DIWA | 60(for averaged models) | 15x | 68.0% |

Table 1: Computational Cost: ERM++ achieves high performance without extensive hyper-parameter searches, instead using reasonable default ones. Even without Long Training (Section 3.1), we're able to achieve SOTA performance on ResNet-50 averaged across TerraIncognita, OfficeHome, PACS, DomainNet and VLCS. Train FLOPs are relative to ERM++w/out *LT*

## 3.3 Weight-Space Regularization

Regularization has long been used to prevent over-fitting of models to training data. Overfitting is an even bigger challenge in DG, because the the source data has a different distribution than the target distribution. One regularization technique is averaging model iterates (Arpit et al., 2021; Cha et al., 2021; Izmailov et al., 2018; Ruppert, 1988; Wortsman et al., 2022a;b; Rame et al., 2022; Li et al., 2022), which improves generalization by converging to flatter minima (Izmailov et al., 2018). Methods can roughly be divided into those which average within a single trajectory (Arpit et al., 2021; Izmailov et al., 2018; Cha et al., 2021), and those between different trajectories originating from a single parent (Li et al., 2022; Wortsman et al., 2022a; Rame et al., 2022).
**Model Parameter Averaging, (*MPA*):** Arpit et al. (2021) revisit a simple method for parameter averaging where simply all iterates are averaged(*MPA*). We verify that **MPA** works in combination with other techniques present in ERM++. In a departure from most of the other improvements explored (wrt. using domain labels), we also experiment with training domain experts to induce model diversity(*SMPA*), but find that this does not result in improved performance over within-trajectory averaging. Therefore *MPA* is part of ERM++, but *SMPA* is not.

## 3.4 ERM++ Computational Cost

ERM++ induces less training cost overhead compared to competing methods, see Table 1. DIWA (Rame et al., 2022) and MIRO (Cha et al., 2022) both use expensive hyper-parameter searches, while we simply use reasonable default ones. Overall, without long training, ERM++ achieves SOTA accuracy with 50% of the training compute of MIRO and 5% of the compute of DIWA (Rame et al., 2022), while retaining the same inference overhead.

## 4 Experimental Settings

We benchmark ERM++ on a diverse set of datasets commonly used for evaluating multi-source DG:
**OfficeHome** (Venkateswara et al., 2017) is a 65-way classification problem depicting everyday objects from 4 domains: art, clipart, product, and real, with a total of 15,588 samples.
**DomainNet** (Peng et al., 2019) is 345-way object classification problem from 6 domains: clipart, infograph, painting, quickdraw, real, and sketch. With a total of 586,575 samples, it is larger than most of the other evaluated datasets in both samples and classes.
**PACS** (Li et al., 2017) is a 7-way object classification problem from 4 domains: art, cartoon, photo, and sketch, with 9,991 samples. It helps verify our method in smaller-scale settings.
**VLCS** (Fang et al., 2013) is a 5-way classification problem from 4 domains: Caltech101, LabelMe, SUN09, and VOC2007. There are 10,729 samples. VLCS is a good test for close OOD; the member datasets are all real photos. The distribution shifts are subtle and simulate real-life scenarios well.
**TerraIncognita** (Beery et al., 2018) is a 10-way classification problem of animals in wildlife cameras, where the 4 domains are different locations. There are 24,788 samples. This represents a realistic use-case where generalization is indeed critical.
**Wilds-FMOW** (Koh et al., 2021; Christie et al., 2018) is a 62-way land-use classification problem, with satellites from 5 regions as different domains. There are 141,696 samples. Wilds-FMOW is a realistic problem different from the above and not focused on objects.

We follow the DomainBed training procedure and add additional components from ERM++. In particular, we use the default hyper-parameters from DomainBed (Gulrajani & Lopez-Paz, 2020),

| | OH | PA | DN | TI | VL | Avg. |
|---|---|---|---|---|---|---|
| IRM (Arjovsky et al., 2019) | $64.3_{\pm2.2}$ | $83.5_{\pm0.8}$ | $33.9_{\pm2.8}$ | $47.6_{\pm0.8}$ | $78.5_{\pm0.5}$ | 61.6 |
| CDANN (Li et al., 2018b) | $65.8_{\pm1.3}$ | $82.6_{\pm0.9}$ | $38.3_{\pm0.3}$ | $45.8_{\pm1.6}$ | $77.5_{\pm0.1}$ | 62.0 |
| DANN (Ganin et al., 2016) | $65.9_{\pm0.6}$ | $83.6_{\pm0.4}$ | $38.3_{\pm0.1}$ | $46.7_{\pm0.5}$ | $78.6_{\pm0.4}$ | 62.6 |
| MTL (Blanchard et al., 2021) | $66.4_{\pm0.5}$ | $84.6_{\pm0.5}$ | $40.6_{\pm0.1}$ | $45.6_{\pm1.2}$ | $77.2_{\pm0.4}$ | 62.9 |
| Mixup (Xu et al., 2020; Yan et al., 2020) | $68.1_{\pm0.3}$ | $84.6_{\pm0.6}$ | $39.2_{\pm0.1}$ | $47.9_{\pm0.8}$ | $77.4_{\pm0.6}$ | 63.4 |
| MLDG (Li et al., 2018a) | $66.8_{\pm0.6}$ | $84.9_{\pm1.0}$ | $41.2_{\pm0.1}$ | $47.7_{\pm0.9}$ | $77.2_{\pm0.4}$ | 63.6 |
| ERM (Vapnik, 1999) | $67.6_{\pm0.2}$ | $84.2_{\pm0.1}$ | $44.0_{\pm0.1}$ | $47.8_{\pm0.6}$ | $77.3_{\pm0.1}$ | 64.2 |
| CORAL (Sun & Saenko, 2016) | $68.7_{\pm0.3}$ | $86.2_{\pm0.3}$ | $41.5_{\pm0.1}$ | $47.6_{\pm1.0}$ | $78.8_{\pm0.6}$ | 64.5 |
| mDSDI (Bui et al., 2021) | $69.2_{\pm0.4}$ | $86.2_{\pm0.2}$ | $42.8_{\pm0.1}$ | $48.1_{\pm1.4}$ | $79.0_{\pm0.3}$ | 65.1 |
| MIRO (Cha et al., 2022) | $70.5_{\pm0.4}$ | $85.4_{\pm0.4}$ | $44.3_{\pm0.2}$ | $50.4_{\pm1.1}$ | $79.0_{\pm0.0}$ | 65.9 |
| SWAD (Cha et al., 2021) | $70.6_{\pm0.2}$ | $88.1_{\pm0.1}$ | $46.5_{\pm0.1}$ | $50.0_{\pm0.3}$ | $79.1_{\pm0.1}$ | 66.9 |
| CORAL + SWAD (Sun & Saenko, 2016) | $71.3_{\pm0.1}$ | $88.3_{\pm0.1}$ | $46.8_{\pm0.0}$ | $51.0_{\pm0.1}$ | $78.9_{\pm0.1}$ | 67.3 |
| DIWA (Rame et al., 2022) | 72.8 | 89.0 | 47.7 | 51.9 | 78.6 | 68.0 |
| MIRO + SWAD (Cha et al., 2022) | $72.4_{\pm0.1}$ | $88.4_{\pm0.1}$ | $47.0_{\pm0.0}$ | $\mathbf{52.9}_{\pm0.2}$ | $\mathbf{79.6}_{\pm0.2}$ | 68.1 |
| ERM++ (Ours) | $\mathbf{74.7}_{\pm0.0}$ | $\mathbf{89.8}_{\pm0.3}$ | $\mathbf{50.8}_{\pm0.0}$ | $51.2_{\pm0.3}$ | $78.0_{\pm0.1}$ | $\mathbf{68.9}$ |

Table 2: **Comparison to recent methods:** Performance of recent methods as reported by (Cha et al., 2022). ERM outperforms almost all prior work, especially when combined with techniques such as SWAD and MIRO. ERM++ outperforms all prior work on average. DIWA does not report confidence intervals.

e.g. , a batch size of 32 (per-domain), a learning rate of 5e-5, a ResNet dropout value of 0, and a weight decay of 0. Unless we specify that the **_Long Training_** component is added, we train models for 15000 steps on DomainNet (following SWAD(Cha et al., 2021)) and 5000 steps for other datasets, which corresponds to a variable number of epochs dependent on dataset size. If **_Long Training_** is used, we extend training by 4x. We train on all source domains except for one, validate the model on held-out data from the sources every 300 steps, and evaluate on the held-out domain.

## 5 RESULTS

| | OH | PA | VL | DN | TI | Avg |
|---|---|---|---|---|---|---|
| MIRO + SWAD | 72.4 | 88.4 | **79.6** | 47.0 | 52.9 | 68.1 |
| DIWA | 72.8 | 89.0 | 78.6 | 47.7 | 52.9 | 68.0 |
| ERM++ | 74.7 | 89.8 | 78.0 | 50.8 | 51.2 | 68.9 |
| DIWA + ERM++ | 75.1 | **90.0** | 78.6 | **51.5** | 51.4 | 69.3 |
| CORAL + ERM++ | 66.9 | 83.8 | 79.3 | 46.2 | 48.1 | 64.9 |
| MIRO + ERM++ | **76.3** | 88.8 | 77.9 | 50.4 | **53.4** | **69.4** |

(a) Without a well-tuned baseline, it can be challenging to determine whether reported improvements are due to the proposed method's effectiveness or simply a result of arbitrary choices. We combine ERM++ with MIRO (Cha et al., 2022), DIWA (Rame et al., 2022), and CORAL (Sun & Saenko, 2016). Both DIWA and MIRO improve performance, validating that DIWA and MIRO are effective methods even when built on top of a stronger baseline. However, CORAL proves to be brittle and does not perform well when combined with ERM++.

| | OH | PA | VL | DN | TI | Avg |
|---|---|---|---|---|---|---|
| ERM | 66.4 | 83.4 | 75.9 | 44.4 | 35.3 | 61.1 |
| ERM++ | **83.4** | **91.1** | **81.5** | **58.8** | **48.3** | **72.6** |

(b) CLIP-ViT-B/16(Radford et al., 2021): With all ERM++ components except *UBN (Unfrozen Batch Norm)*, ERM++ outperforms ERM by over 10%, showing the generality of ERM++ for different architectures. Furthermore, ERM training of the ViT does not outperform ResNet-50, meaning that standard ERM cannot leverage the much stronger pre-training of CLIP. In contrast, ERM++ allows us to effectively use ViT's.

Table 3: ERM++ in combination with other methods **(a.)** and compared to standard ERM on a CLIP pretrained ViT-B/16 **(b.)**

Table 2 compares ERM++ to prior work, where we outperform the state-of-the-art across five DomainBed datasets by an average of 1%. The single largest gain was on DomainNet (3% gain), with OfficeHome and PACS obtaining still substantial gains of 1.5-2%. Table 3(a) demonstrates our training procedure's ability to generalize, where we combine our approach with several of the

| ERM++ Components (#7 is full ERM++) | | | | | | | OfficeHome | PACS | VLCS | DomNet | TerraInc | Avg. |
|---|---|---|---|---|---|---|---|---|---|---|---|---|
| # | MPA | FD | LT | WS | ES | S. Init | UBN | 15K | 10K | 11K | 590K | 25K | |
| 1 | ✗ | ✗ | ✗ | ✗ | ✗ | ✗ | ✓ | $67.1_{\pm0.2}$ | $85.1_{\pm0.3}$ | $76.9_{\pm0.6}$ | $44.1_{\pm0.15}$ | $45.2_{\pm0.6}$ | 63.7 |
| 2 | ✓ | ✗ | ✗ | ✗ | ✗ | ✗ | ✓ | $70.2_{\pm0.3}$ | $85.7_{\pm0.2}$ | $78.5_{\pm0.3}$ | $46.4_{\pm0.0}$ | $49.4_{\pm0.4}$ | 66.0 |
| 3 | ✓ | ✓ | ✗ | ✗ | ✗ | ✗ | ✓ | $71.5_{\pm0.1}$ | $87.3_{\pm0.2}$ | $77.4_{\pm0.1}$ | $46.8_{\pm0.0}$ | $49.8_{\pm0.5}$ | 66.5 |
| 4 | ✓ | ✓ | ✓ | ✗ | ✗ | ✗ | ✓ | $71.7_{\pm0.1}$ | $88.7_{\pm0.2}$ | $76.9_{\pm0.1}$ | $48.3_{\pm0.0}$ | $49.6_{\pm0.4}$ | 67.0 |
| 5 | ✓ | ✓ | ✓ | ✓ | ✗ | ✗ | ✓ | $72.6_{\pm0.1}$ | $88.8_{\pm0.1}$ | $77.0_{\pm0.1}$ | $48.6_{\pm0.0}$ | $49.3_{\pm0.3}$ | 67.3 |
| 6 | ✓ | ✓ | ✓ | ✓ | ✓ | ✗ | ✓ | $72.6_{\pm0.1}$ | $88.8_{\pm0.1}$ | $\mathbf{78.7}_{\pm0.0}$ | $48.6_{\pm0.0}$ | $49.2_{\pm0.3}$ | 67.6 |
| 7 | ✓ | ✓ | ✓ | ✓ | ✓ | ✓ | ✓ | $\mathbf{74.7}_{\pm0.0}$ | $89.8_{\pm0.3}$ | $78.0_{\pm0.1}$ | $\mathbf{50.8}_{\pm0.0}$ | $\mathbf{51.2}_{\pm0.3}$ | **68.9** |
| 8 | ✓ | ✓ | ✗ | ✓ | ✓ | ✓ | ✓ | $74.6_{\pm0.1}$ | $87.9_{\pm0.2}$ | $78.6_{\pm0.1}$ | $49.8_{\pm0.0}$ | $51.1_{\pm0.8}$ | 68.4 |
| 9 | ✓ | ✓ | ✓ | ✓ | ✓ | ✓ | ✗ | $\mathbf{74.7}_{\pm0.2}$ | $\mathbf{90.1}_{\pm0.0}$ | $78.6_{\pm0.1}$ | $49.9_{\pm0.0}$ | $49.0_{\pm0.4}$ | 68.3 |

Table 4: We present the overall ablation for ERM++. ERM++ corresponds to experiment 7. (1) ERM (Gulrajani & Lopez-Paz, 2020) baseline with unfrozen BN. (2) MPA: Model parameter averaging, which uniformly improves results. (3) FD: training on the full data. (4) LT: Training for 4x longer, which ensures convergence improves performance by an additional half percent. (5) WS: Warm-starting the classification layer especially improves OfficeHome. (6) ES: Splitting off validation data to find a training length yields substantial gains. (7) S.Init: Initializing the initial parameters to those trained with AugMix brings performance to state of the art. (8) Removing LT from (7) still results in state-of-the-art performance with half of the training cost of MIRO. (9) UBN: When we freeze the BN parameters, we see that performance substantially degrades.

highest performing methods in prior work (DIWA (Rame et al., 2022), MIRO (Cha et al., 2022), and CORALSun & Saenko (2016)). We find that our approach is able to boost the performance of DIWA and MIRO by over 1%, while CORAL decreases performance by several percent. This validates that DIWA and MIRO are effective methods even when built on top of a stronger baseline, while CORAL is more brittle. It also demonstrates the importance tuning the baseline before drawing conclusions. Finally, in Table 3(b), we show that ERM++ components also bring a massive benefit to the DG capabilities of CLIP-pretrained VIT-B/16 models, demonstrating the generality of ERM++. We provide a detailed analysis of each component below.

## 5.1 DATA UTILIZATION

**Using the full data (FD):** The most common ERM (Gulrajani & Lopez-Paz, 2020) implementation splits off 80% of the source domains for training, and keeps the remaining 20% for hyper-parameter validation and checkpoint selection. By comparing Table 4 in experiments 2 and 3, we show that training on the full data improves over checkpoint selection on a validation set on all datasets except for VLCS. Early Stopping (*ES*) below helps us recover VLCS performance.

**Long training (LT):** Prior work has shown that training to proper convergence can have large impacts on transfer learning performance (Chen et al., 2020). To explore this setting for DG, we extended training by 4x for each dataset. In other words, DomainNet models are trained for 60K steps while the other datasets are trained for 20K steps. This training length is one where we observe source validation accuracies start to saturate for most datasets (see Appendix B.3). We present the results in Table 4, experiment 4. Training for longer, on average, increases performance by 0.5%.

**Early Stopping (ES):** Although the training pieces presented so far improve DG performance on the datasets considered on average, one consistent pattern is that VLCS performance degrades in experiments number 3 (*FD*), 4 (*LT*). This suggests that VLCS is a dataset which is prone to overfitting. We observe that this is true even on a validation set constructed from the source domains. Therefore, we propose an additional step where we use 20% validation splits in order to search for the proper number of training steps, and then retrain using the full data. In Table 4, Experiment 6, we see this dramatically improves VLCS performance w/out affecting other datasets.

## 5.2 PRETRAINED MODEL WEIGHT USAGE

**Warmstart (WS):** In Table 4, we compare to training using a random initialization for the new classification layer (Experiment 4) or by using Warmstart (Experiment 5). We find **WS** provides

| | OffHome | PACS | VLCS | DomNet | TerraInco | Avg | ImgNet |
|---|---|---|---|---|---|---|---|
| TorchVision Weights | 72.2 | 85.9 | 78.5 | 46.9 | 49.7 | 66.6 | 76.1 |
| AugMix (Hendrycks et al., 2020) | 74.6 | **87.9** | 78.6 | **49.8** | **51.0** | **68.4** | 79.0 |
| Meal V2 (Shen & Savvides, 2020) | **75.5** | 86.7 | **79.1** | 49.5 | 50.9 | 68.3 | **80.7** |
| ResNet A1 (Wightman et al., 2021) | 70.8 | 82.8 | 77.7 | 43.0 | 37.3 | 62.3 | 80.4 |

Table 5: **Top-1 Accuracy with different ResNet-50 initialization**: We investigate initialization weights from different pre-training procedures. The differences between different initializations are very substantial, up to about 6%. Interestingly, improved ImageNet accuracy does not strongly correlate with improved performance. In fact, the strongest initialization is from AugMix pretrained weights, with an ImageNet validation accuracy 2% less than the strongest model. Additionally, MealV2 is a distilled model from a very strong ensemble, where the student is initialized to AugMix weights. The distillation process doesn't improve generalization performance overall, improving over AugMix only in domains which resemble ImageNet. This suggests that the distillation process effectively matches the student to the teacher over the data used in the distillation process, but not elsewhere.

| | Painting | Clipart | Info | Real | Quickdraw | Sketch | Avg |
|---|---|---|---|---|---|---|---|
| Aug(Hendrycks et al., 2020) | **57.3** | **68.8** | **25.6** | 70.2 | **17.1** | **59.8** | **49.8** |
| MV2(Shen & Savvides, 2020) | **57.3** | 68.5 | 25.4 | **70.9** | 16.1 | 59.0 | 49.5 |

Table 6: **Model distillation's effect on DG:** We look at the per-domain accuracy on DomainNet, comparing Augmix training (Aug) and MealV2 (MV2). MealV2 is a method used to distill a large ensemble into a student ResNet-50, where the student is initialized to AugMix weights. We can see that the distillation process, while dramatically improving ImageNet performance, only slightly changes DG performance. In particular, generalization gets slightly worse for all domains except for (R)eal, which is the most similar to ImageNet. This is surprising, since it has been shown that both ensembles Arpit et al. (2021) and larger models Angarano et al. (2022) improve DG performance.

a small but consistent boost on average across datasets. We find this is likely due to a decrease in overfitting to the source domains. We verify that WS has a regularization effect by measuring the L2 distance of the final model from initialization (the pre-trained model) and find that the trained weights were more than twice as far without using **WS** (58.1 with and 122.5 w/o).

**Unfreezing the Batchnorm (*UBN*):** BatchNorm is commonly frozen in current DG recipes for reasons not well justified. However, we find that frozen batch normalization leads to quick overfitting in the long-training regime. In Figure 4 we can see that frozen batch normalization results in overfitting. In contrast, without frozen batch normalization this is not an issue. As seen in Table 4, Experiment 9, freezing BN also results in lower performance. It can be concluded that unfrozen BatchNorm, gives an effective regularization effect by randomizing shifting and scaling of features.

**Stronger initializations (*S. Init*):** One of the key components of the standard DG training scheme is initializing the model parameters with a pre-trained model. The effect of the strong initialization for our model is shown in Table 4, experiment 7, where we achieve a 1% boost an average. However, selecting a model takes care. Table 5 compares ResNet-50 models of varying ImageNet performance described in Section 3.2. We summarize our findings below:

- Stronger ImageNet performance does not necessarily correspond to better DG performance. In particular, both the ResNet-50 A1 and Meal V2 weights achieves much better ImageNet Top-1 Accuracy than the standard TorchVision weights, but do not achieve the best DG performance. However, the overall consistency of the AugMix weights across all 5 datasets makes it a reasonable choice.

- Model Distillation, which strongly improves source accuracy, does not increase overall DG performance. Meal-V2 is a distillation of the ensemble if two very strong ImageNet models into a ResNet-50. Interestingly, the student in Meal-V2 is initialized with the same AugMix trained network as we do in our experiments. Therefore, the differences in performance can be strictly attributed to the effects of model distillation. Looking at the results in more detail, as in Table 6, we can see that performance on ImageNet-like domains improves while performance on other domains suffers. This suggests that the distillation process effectively matches the student to the teacher over the data used in the distillation process, but not elsewhere.

|  | R0 | R1 | R2 | R3 | R4 | Avg |
|---|---|---|---|---|---|---|
| ERM | 34.9 | 47.1 | 38.6 | 43.8 | 53.8 | 43.6 |
| ERM++ | **41.5** | 50.3 | 40.5 | 50.4 | 57.7 | 48.1 |
| - Strong Init | 39.2 | 50.1 | 39.5 | 49.5 | 58.7 | 47.4 |
| - WS | 41.3 | **50.4** | **41.0** | **50.6** | **59.3** | **48.5** |
| - UBN | 39.6 | 49.1 | 38.9 | 49.1 | 58.1 | 47.0 |

(a) **WILDS-FMOW Top-1 Accuracy:** We show that ERM++ outperforms ERM on this on the challenging WILDS-FMOW classification dataset. We also ablate several components of ERM++. UBN (Unfrozen Batch Norm) and Strong Init (from Augmix) improve performance, while surprisingly WS (warmstart) decreases performance in this particular scenario. We emphasize that ERM++ overall improves over ERM(Gulrajani & Lopez-Paz, 2020).

|  | P | I | Q | S | R | C | Avg |
|---|---|---|---|---|---|---|---|
| ERM | 51.1 | 21.2 | 13.9 | 52.0 | 63.7 | 63.0 | 44.1 |
| SMPA | 52.9 | **27.2** | 14.3 | 51.3 | 65.6 | 65.2 | 46.1 |
| MPA | **55.2** | 24.0 | **16.7** | **57.4** | **67.0** | **67.49** | **48.0** |

(b) **Weight Space Regularization:** We show experiments different types of parameter averaging for weight regularization on DomainNet. **SMPA** is a specialized model parameter averaging, where we average parameters of domain specialists, while **MPA** averages parameters within a single training trajectory. While both **MPA** and **SMPA** outperform ERM, **MPA** outperforms **SMPA**.

Table 7: (a.) Compares ERM and ERM++ on Wilds-FMOW, while in (b.) we show the effect of model parameter averaging.

- AugMix is a model trained with generalization to synthetic corruptions as a goal and results in a very strong DG performance. Therefore, while ImageNet Top-1 accuracy is not a good indicator for DG performance, investigating the correlation between synthetic corruption performance and DG performance is promising.

## 5.3 WEIGHT SPACE REGULARIZATION

**Generalist Model Parameter Averaging (*MPA*):** We confirm that regularizing model parameters by averaging iterates is an important tool in improving DG performance; in Table 4 (Experiments 1 and 2) we compare models trained with and without parameter averaging across timesteps. Specifically, we average the parameters of all training steps after an initial burn-in period of 100 steps. We confirm that such model parameter averaging consistently and substantially improves DG.

**Specialist Model Parameter Averaging (*SMPA*):** We also explored a setting where instead of averaging model weights, we attempt to include diversity between the models being averaged as this has been shown to boost performance (Rame et al., 2022). Following (Li et al., 2022), we first train a generalist model on all source domains for 5 epochs, then train specialist models for 5 epochs, before averaging parameters. Results on the DomainNet dataset are reported in Table 3b. Although averaging specialists improves over ERM, it does not improve over averaging iterates of a generalist.

## 5.4 GENERALIZING BEYOND WEB-SCRAPED DATASETS

We have demonstrated that ERM++ is a highly effective recipe for DG on several datasets: Office-Home, PACS, DomainNet, and TerraIncognita. These datasets are diverse and represent a strong evaluation of ERM++. However, (Fang et al., 2023) show that on datasets not consisting of web-scraped data, the correlation between ImageNet performance and transfer performance is quite weak. To verify that this is not the case for ERM++, we perform an ablation study on WILDS-FMOW, a land-use classification dataset, and see that ERM++ substantially improves over ERM (Table 3a).

## 6 CONCLUSION

This paper develops a strong baseline, ERM++, that can be used to improve the performance of DG models. By identifying several techniques for enhancing ERM, our approach achieves significant gains in DG performance, reporting a 1% average boost over the state-of-the-art on the challenging DomainBed evaluation datasets and demonstrating efficacy in realistic deployment scenarios on WILDS-FMOW. We find that ERM++ can also boost the performance of state-of-the-art methods, and that it improves ViT models pretrained on CLIP by over 10%. Our results highlight the importance of improving the training procedure for better DG performance and provide a strong baseline for future research. ERM++ opens up opportunities for exploring additional techniques to further improve DG performance.

## 7 ETHICS STATEMENT

In general, methods which generalize well to new domains are *more* likely to results in fair,accurate, and ethical systems. Nevertheless, the assumption that a trained model will *always* generalize to new data is dangerous, and we caution readers that much improved robustness on unseen domains is still not perfect, or even good, robustness.

## 8 REPRODUCIBILITY STATEMENT

We release the code in the provided zip file, and provide training details in Appendix E.

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

## A    APPENDIX

## B    ADDITIONAL RESULTS

### B.1    REGULARIZATION EFFECTS OF UNFREEZING BATCH NORM.

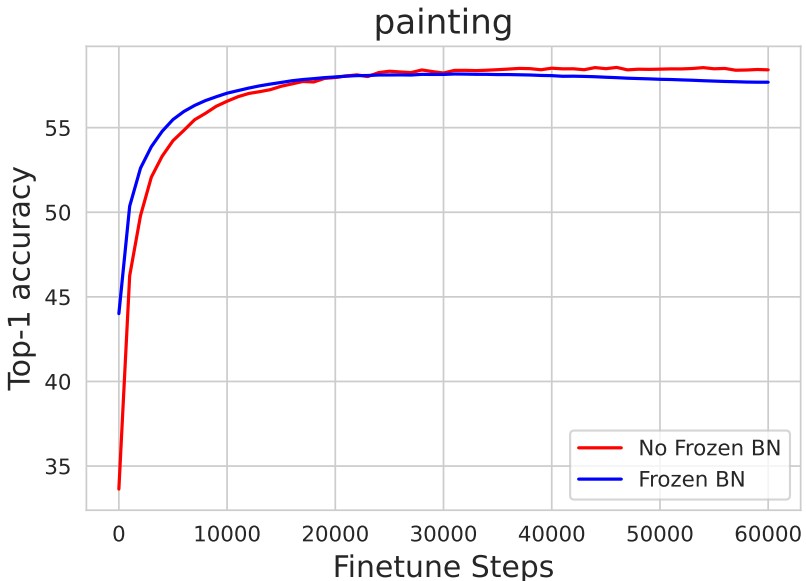

Figure 4: **Unfreezing Batchnorm:** Here we show the test curves of the fine-tuning on the held-out painting domain of DomainNet. With frozen BatchNorm, training is faster but it overfits.

### B.2    PER-DATASET DETAILS

In Tables 8 (OfficeHome), 9 (DomainNet), 10 (VLCS), 11 (TerraIncognita), 12 (PACS), we expand results for the datasets and report accuracies for each held-out domain. We compare ERM++ with reported performances of ERM (Gulrajani & Lopez-Paz, 2020), DIWA (Rame et al., 2022), SWAD, (Cha et al., 2021), and MIRO  (Cha et al., 2022). ERM + SWAD + MIRO and DIWA are the current SOTA for ResNet-50 models for this set of datasets. Overall trends include ERM++ being especially effective at sketch-like domains, indicating a lowered texture bias. On the sketch and clipart domains in DomainNet, ERM++ outperforms prior best performance by over 4%. When we additionally combine MIRO with ERM++, we see much improved performance on OfficeHome and TerraIncognita without much affecting the performance on the other datasets.

### B.3    VALIDATION-SET ACCURACY CURVES

In Figures 12,13,14,15, and 16, we provide source-validation accuracies for each of the 5 datasets, for the number of steps corresponding to *long training*, which is 20000 steps for most datasets except for the larger DomainNet, which is 60000 steps. As one can see, at this point, validation accuracy is saturated for most domains in most datasets, so this training length is reasonable. Prior training lengths are denoted as red vertical lines in these figures, and one can see that for many datasets this is not a sufficient training length. As we describe in Section 5.1 of the main paper, this improves performance by 0.5% on average.

## C    DATASET VISUALIZATIONS

In Figures 5 (OfficeHome), 6 (DomainNet), 7 (VLCS), 8 (TerraIncognita), 9 (PACS), 10 (FMoW) we show samples of a few classes from each of the datasets, and each domain. As one can see, both

| | art | clipart | product | real | avg |
|---|---|---|---|---|---|
| ERM Gulrajani & Lopez-Paz (2020) | 63.1 | 51.9 | 77.2 | 78.1 | 67.6 |
| ERM + SWAD Cha et al. (2021) | 66.1 | 57.7 | 78.4 | 80.2 | 70.6 |
| DIWA Rame et al. (2022) | 69.2 | 59 | 81.7 | 82.2 | 72.8 |
| ERM + MIRO + SWAD Cha et al. (2022) | - | - | - | - | 72.4 |
| ERM++ | 70.7 | **62.2** | 81.8 | 84.0 | 74.7 |
| ERM++ + MIRO | **74.0** | 61.5 | **83.8** | **85.7** | **76.3** |

Table 8: **OfficeHome:** Per-domain top-1 accuracy against reported results of recent top-performing methods SWAD, DIWA, and MIRO. Cha et al. (2022) does not report per-domain performance for MIRO, so we only show average for that case. DIWA doesn't report standard errors. ERM++ not only greatly increases performance relative to SWAD, DIWA, and MIRO but also reduce variance between runs. The largest gains are on the held-out domain with the largest domain shift(clipart), illustrating the ability of ERM++ to improve performance on difficult DG tasks.

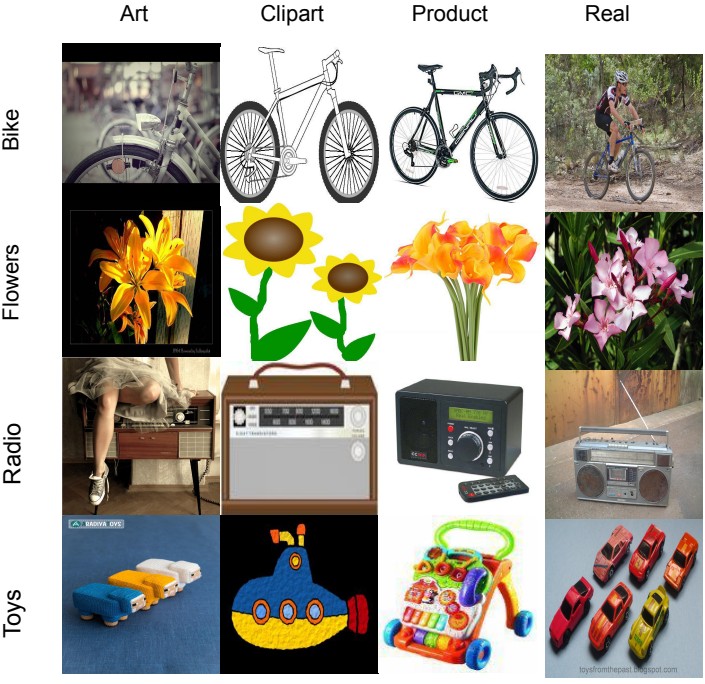

Figure 5: **OfficeHome**:Samples from the OfficeHome dataset, from each domain and selected classes. The dataset focuses on household objects. The domain shifts are in low-level style mostly, and there is little spatial bias.

the datasets and distribution shifts are quite diverse, highlighting the flexibility of our method. We present some key attributes of the datasets below.

**OfficeHome (Venkateswara et al., 2017)** Figure 5. This dataset focuses on household objects. The domain shifts are in low-level style mostly, and there is little spatial bias.

| | painting | clipart | info | real | quickdraw | sketch | avg |
|---|---|---|---|---|---|---|---|
| ERM Gulrajani & Lopez-Paz (2020) | 50.1 | 63.0 | 21.2 | 63.7 | 13.9 | 52.9 | 44.0 |
| ERM + SWAD Cha et al. (2021) | 53.5 | 66.0 | 22.4 | 65.8 | 16.1 | 55.5 | 46.5 |
| DIWA Rame et al. (2022) | 55.4 | 66.2 | 23.3 | 68.7 | 16.5 | 56 | 47.7 |
| ERM + MIRO + SWAD Cha et al. (2022) | - | - | - | - | - | - | 47.0 |
| ERM++ | 58.4 | **71.5** | 26.2 | 70.7 | **17.3** | **60.5** | **50.8** |
| ERM++ + MIRO | **58.5** | 71.0 | **26.5** | **71.1** | 15.9 | 59.5 | 50.4 |

Table 9: **DomainNet:** Per-domain top-1 accuracy against reported results of recent top-performing methods SWAD, DIWA, and MIRO. Cha et al. (2022) does not per-domain performance for MIRO, so we only show average for that case. DIWA doesn't report standard errors. ERM++ not only greatly increases performance relative to SWAD, DIWA, and MIRO but also reduce variance between runs. Similar to results on OfficeHome (Table 8), the largest performance gains(of larger than 4%) are on domains very different from the source domain(clipart and sketch). This suggests ERM++ is less sensitive to texture bias than ERM Gulrajani & Lopez-Paz (2020). The bias of MIRO to the pre-trained weights manifests in slightly higher performance on close to ImageNet domains like real when combined with ERM++, at the slight expense of performance on other domains.

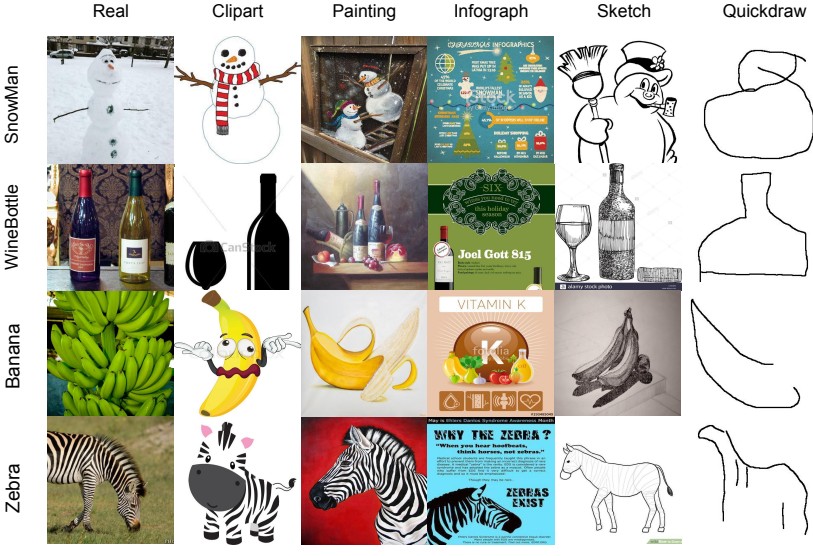

Figure 6: **DomainNet:** Samples from the DomainNet dataset. While the real domain is quite similar to what one might expect in ImageNet, the distribution shifts are quite substantial in other domains. Quickdraw and Infograph are particularly challenging, so the 1-3% gains of ERM++ on these domains is meaningful (Table 9). While most domains contain primarily shifts in low level statistics (for example, real to painting), Infograph also has many non-centered objects.

**DomainNet (Peng et al., 2019)** Figure 6. While the real domain is quite similar to what one might expect in ImageNet, the distribution shifts are quite substantial in other domains. Quickdraw and Infograph are particularly challenging, so the 1-3% gains of ERM++ on these domains is meaningful (Table 9).

|  | caltech101 | labelme | sun09 | voc2007 | avg |
|---|---|---|---|---|---|
| ERM Gulrajani & Lopez-Paz (2020) | 97.7 | **64.3** | 73.4 | 74.6 | 77.3 |
| ERM + SWAD Cha et al. (2021) | 98.8 | 63.3 | **75.3** | **79.2** | 79.1 |
| DIWA Rame et al. (2022) | **98.9** | 62.4 | 73.9 | 78.9 | 78.6 |
| ERM + MIRO + SWAD Cha et al. (2021) | - | - | - | - | **79.6** |
| ERM++ | 98.7 | 63.2 | 71.6 | 78.7 | 78.0 |
| ERM++ + MIRO | 99.0 | 62.4 | 71.8 | 78.3 | 77.9 |

Table 10: **VLCS:** Per-domain top-1 accuracy against reported results of recent top-performing methods SWAD, DIWA, and MIRO. Cha et al. (2022) does not per-domain performance for MIRO, so we only show average for that case. DIWA doesn't report standard errors. Although overall performance on VLCS is lower than competing methods, we can see that this drop primarily comes from lower performance on sun09. Furthermore, there are many ambiguous images in the LabelMe domain (see Figure 11), raising questions about the usefulness of trying to train on this domain.

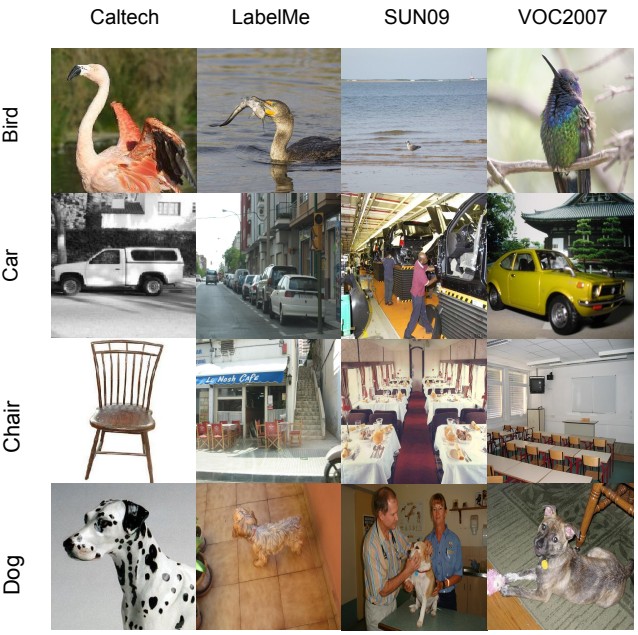

Figure 7: **VLCS:** The low-level statistics are quite similar between domains, however spatial biases differ between domains. Caltetch objects are quite centered, while other domains do not have this trait. For example the LabelMe domain has cars along the side of the image, and there are many chairs in the VOC2007 domain. Furthermore, in some cases the size of the objects differs dramatically. Finally, there are many ambiguous images in the LabelMe domain (see Figure 11), raising questions about the usefulness of trying to train on this domain.

**VLCS (Fang et al., 2013):** Figure 7. Low-level statistics are quite similar between domains in this dataset, however spatial biases differ between domains. For example, Caltetch objects are quite centered, while other domains do not have this trait. For example the LabelMe domain has cars along the side of the image, and there are many chairs in the VOC2007 domain. Furthermore, in some cases the size of the objects differs dramatically. Lastly, there are many ambiguous images

| | Loc. 100 | Loc. 38 | Loc. 43 | Loc. 46 | Average |
|---|---|---|---|---|---|
| ERM Gulrajani & Lopez-Paz (2020) | 54.3 | 42.5 | 55.6 | 38.8 | 47.8 |
| ERM + SWAD Cha et al. (2021) | 55.4 | 44.9 | 59.7 | 39.9 | 50.0 |
| DIWA Rame et al. (2022) | **57.2** | 50.1 | 60.3 | 39.8 | 51.9 |
| ERM + MIRO + SWAD Cha et al. (2022) | - | - | - | - | 52.9 |
| ERM++ | 48.3 | **50.7** | **61.8** | **43.9** | 51.2 |
| ERM++ + MIRO | **60.81** | 48.8 | 61.1 | 42.7 | **53.4** |

Table 11: **TerraIncognita:** Per-domain top-1 accuracy against reported results of recent top-performing methods SWAD, DIWA, and MIRO. Cha et al. (2022) does not per-domain performance for MIRO, so we only show average for that case. DIWA doesn't report standard errors. ERM++ outperforms other methods on 3 out of 4 held out domains despite slightly underperforming on average. However, we point out that ERM++ w/MIRO outperforms both DIWA and MIRO, and improves ERM++ by a further 2%.

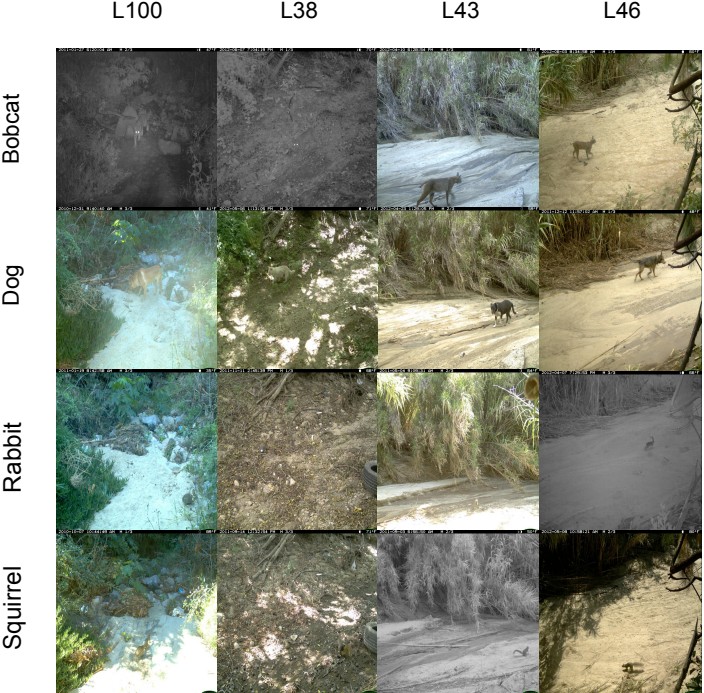

Figure 8: **TerraIncognita**: Samples from the TerraIncognita dataset, from each domain and selected classes. The background stays consistent, and the animal object frequently takes up a small portion of the frame. At night the images are black-and-white. This dataset matches realistic deployment scenarios well.

in the LabelMe domain (see Figure 11), raising questions about the validity of trying to improve performance on this dataset.

|  | art_painting | cartoon | photo | sketch | avg |
|---|---|---|---|---|---|
| ERM Gulrajani & Lopez-Paz (2020) | 84.7 | 80.8 | 97.2 | 79.3 | 84.2 |
| ERM + SWAD Cha et al. (2021) | 89.3 | 83.4 | 97.3 | 82.5 | 88.1 |
| DIWA Rame et al. (2022) | 90.6 | 83.4 | 98.2 | 83.8 | 89 |
| ERM + MIRO + SWAD Cha et al. (2022) | - | - | - | - | 88.4 |
| ERM++ | **90.6** | 83.7 | 98.1 | **86.6** | **89.8** |
| ERM++ + MIRO | 90.2 | **83.8** | **98.6** | 82.4 | 88.8 |

Table 12: **PACS:** Per-domain top-1 accuracy against reported results of recent top-performing methods SWAD, DIWA, and MIRO. Cha et al. (2022) does not per-domain performance for MIRO, so we only show average for that case. DIWA doesn't report standard errors. ERM++ leads to substantial improvement over prior work. As in other dataset (OfficeHome, DomainNet), large performance gains are made on the sketch domain.

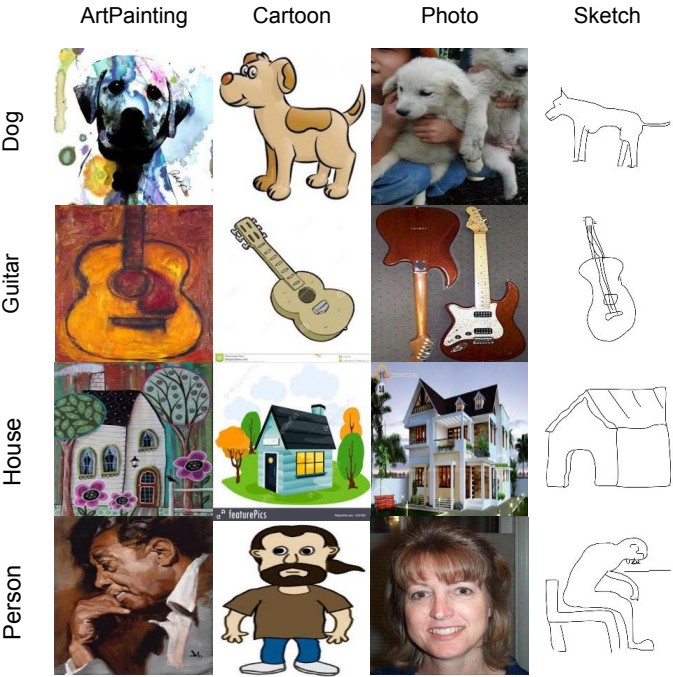

Figure 9: **PACS:** Samples from the PACS dataset, from each domain and selected classes. The subjects tend to be centered, and the sketches are more realistic than the quickdraw setting in Domain-Net. Though the domians are similar to that of DomainNet, PACS has fewer than 10000 samples compared to 586000 of DomainNet. Therefore PACS tests the capabilities of ERM++ on smaller data.

**TerraIncognita (Beery et al., 2018)**: Figure 8 The background stays consistent, and the animal object frequently takes up a small portion of the frame. At night the images are black-and-white. This is a very realistic dataset, on which is good to test.

**PACS (Li et al., 2017)** Figure 9. The subjects tend to be centered, and the sketches are more realistic than the quickdraw setting in DomainNet. Though the domains are similar to that of DomainNet,

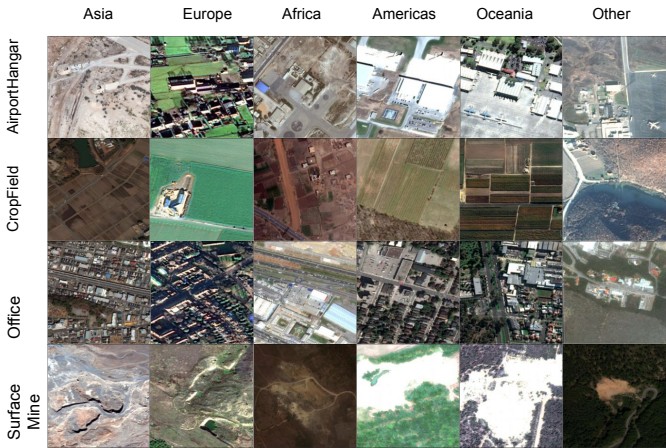

Figure 10: **FMoW:**Samples from the TerraIncognita dataset, from each domain and selected classes. The images differ in region but also in resolution and scale. The distribution shift between FMoW and the pretraining data is large, therefore FmoW represents the ability of ERM++ to perform on non web-scraped data (see Section 5.4 of the main paper).

PACS has fewer than 10000 samples compared to 586000 of DomainNet. Therefore PACS tests the capabilities of ERM++ on smaller data.

**FMoW**: Figure 10. The images differ in region but also in resolution and scale. The distribution shift between FMoW and the pretraining data is large, therefore FmoW represents the ability of ERM++ to perform on non web-scraped data (see Section 5.4 of the main paper).

## D    RUNTIME COMPARISONS

As discussed in the main paper Section 3.4; ERM++ achieves higher predictive performance than competing methods MIRO (Cha et al., 2022) and DIWA (Rame et al., 2022) despite lower computational cost for training. The reason is reduced cost of hyper-parameter search; we use fixed hyper-parameters, borrowed from the DomainBed framework, (see Section E.2 for more details ) while DIWA averages 20-60 models and MIRO search for *4 $\lambda$* weight regularization values in each experiment. Assuming the worst case scenario of training two full passes (one on validation data for number of training steps for *Early Stopping*, and one on full training data with validation data folded in *Full Data*), and the same number of training steps as MIRO; ERM++ costs $\frac{1}{2}$ that of MIRO while obtaining better performance. In particular, this configuration represents Experiment 8 in Table 3 of the main paper.

For each forward step MIRO there is an additional forward pass of the data through the model which is absent in ERM++. On the other hand, ERM++ does take a forward pass through the running average model to update batch normalization statistics, which is not done in former methods. This means that each forward pass is compute-equivalent for ERM++ and MIRO, for a given architecture.

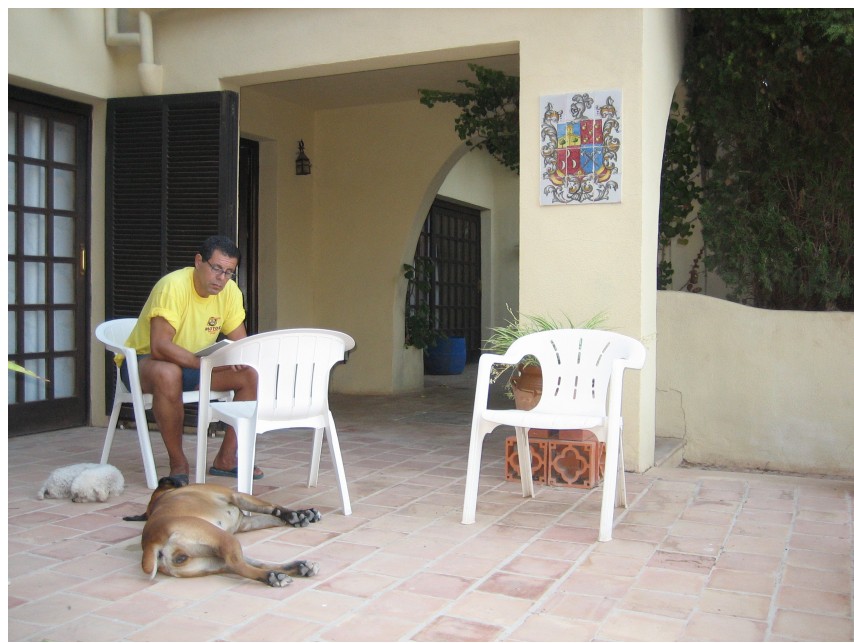

Figure 11: **Sample from LabelMe Domain in VLCS:** Is this a dog, person, or chair? Many samples in the LabelMe domain of VLCS are ambigrous but assigned a label (in this case, dog). This raises questions about the usefulness of training on this domain.

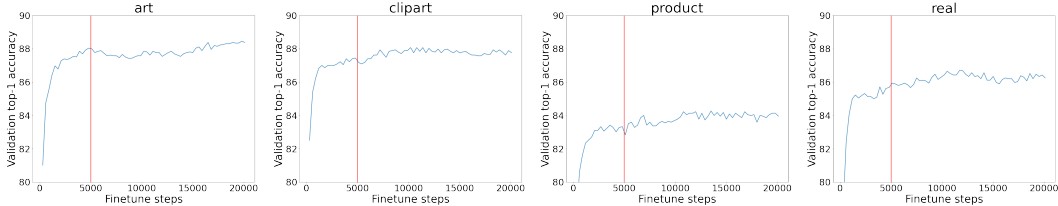

Figure 12: **OfficeHome:** Source validation accuracies. The validation accuracy saturates by 20000 steps, which corresponds to number of steps in *Long Training*(Section 5.1 of the main paper). Training length used in prior works is denoted as a red line, and the training is not yet converged.

# E   REPRODUCIBILITY

We provide code in a zip file along with this supplementary, and will open-source the code upon acceptance.

## E.1   INFRASTRUCTURE

We train on a heterogeneous cluster, primarily on NVIDIA A6000 GPU's. Each experiment is conducted on a single GPU with 4 CPUs. A single run could range from 12-48 hours, depending on number of steps trained.

## E.2   TRAINING DETAILS

We follow the DomainBed (Gulrajani & Lopez-Paz, 2020) training procedure and add additional components from ERM++. In particular, we use the default hyper-parameters from DomainBed (Gulrajani & Lopez-Paz, 2020), e.g. , a batch size of 32 (per-domain), a learning rate of 5e-5, a ResNet dropout value of 0, and a weight decay of 0. We use the ADAM optimizer (Kingma & Ba, 2014) optimizer with $\beta$ and $\epsilon$ values set default values from Pytorch 1.12. Unless we specify that the "Long Training" component is added, we train models for 15000 steps on DomainNet (following

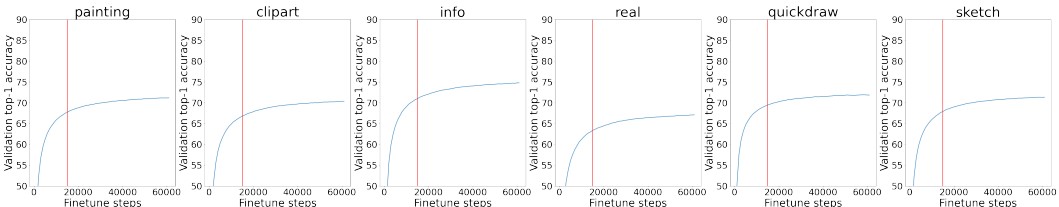

Figure 13: **PACS**: Source validation accuracies. The validation accuracy saturates by 20000 steps, which corresponds to number of steps in *Long Training*(Section 5.1 of the main paper).Training length used in prior works is denoted as a red line, and the training is not yet converged.

Figure 14: **DomainNet:** Source validation accuracies. The validation accuracy saturates by 60000 steps, which corresponds to number of steps in *Long Training*(Section 5.1 of the main paper). Training length used in prior works is denoted as a red line, and the training is not yet converged.

SWAD(Cha et al., 2021)) and 5000 steps for other datasets, which corresponds to a variable number of epochs dependent on dataset size. If Long Training is used, we extend training by 4x. We train on all source domains except for one, validate the model on held-out data from the sources every 300 steps(20% of the source data), and evaluate on the held-out domain. If using *Full Data* we retrain using the full data. We use the same data augmentation techniques as ERM (Gulrajani & Lopez-Paz, 2020). An annotated algorithm is provided in Algorithm 1.

**Model Parameter Averaging details:** If we use Model Parameter Averaging( *MPA*), we begin to keep a running average at the 100th step. If we additionally use warm-start, we only optimize the classification head for the first 500 steps, and start *MPA* 100 steps after that. For the Specialist Model Parameter Averaging(*SMPA*) experiments (Table 6 of main paper), we first train a generalist model for 15000 steps , then train an independent model for each domain for another 1500 steps. At the end, we average parameters and re-compute batch norm running statistics. This recomputing of BN stats makes sure the averaged model has accurately computed batch norm statistics which may not be a simple average of experts, due to the non-linearity of neural nets.

**Batch Normalization details:** With unfrozen batch normalization( *UBN*), we update the evaluation model BN statistics by averaging the model iterates first (from *MPA*), then then forward propagating the current batch at each step through the evaluation model. In this way, the BN running statistics and model used for inference match.

**Sources of pre-trained weights:** We use torchvision 0.13.1 for vanilla ResNet-50 initialization. For augmix and ResNet-A1 initialized weights, we leverage TIMM (Wightman, 2019) [1] [2] .

**A note on hyper-parameter search:** In this work, we focus on methodological improvements that do not depend on expensive hyper-parameter tuning, and as a result we use default learning rate, weight decay, etc. We demonstrate state-of-the-art performance despite this, and greatly reduce the computational cost of training as a result. However, we believe there is substantial headroom for improvement with further hyper-parameter tuning.

---

[1]Augmix Weights :`https://github.com/rwightman/pytorch-image-models/releases/download/v0.1-weights/resnet50_ram-a26f946b.pth`
[2]ResNet-A1 Weights :`https://github.com/rwightman/pytorch-image-models/releases/download/v0.1-rsb-weights/resnet50_a1_0-14fe96d1.pth`

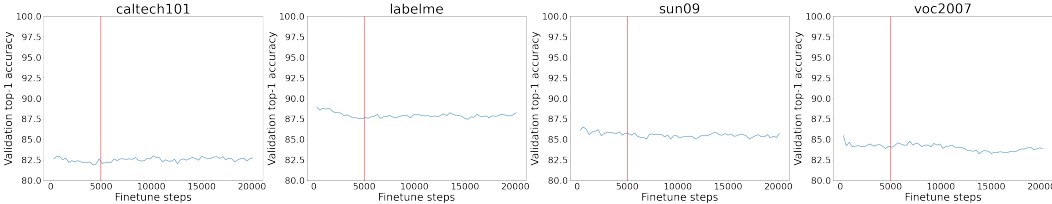

Figure 15: **VLCS:** Source validation accuracies. The validation accuracy saturates by 20000 steps, which corresponds to number of steps in *Long Training*(Section 5.1 of the main paper).Training length used in prior works is denoted as a red line. In the case of VLCS, it seems like longer training is not so helpful, and this is reflected in our ablations (Table 4)
.

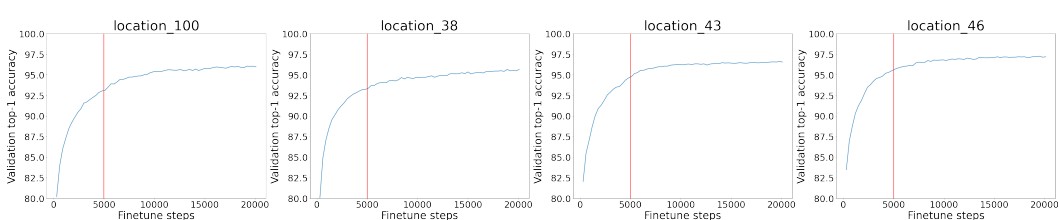

Figure 16: **TerraIncognita:** Source validation accuracies. The validation accuracy saturates by 20000 steps, which corresponds to number of steps in *Long Training*(Section 5.1 of the main paper). Training length used in prior works is denoted as a red line, and the training is not yet converged.

**MIRO Implementation:** We directly follow the MIRO implementation and borrow the lambda weights values from (Cha et al., 2022) when we combine MIRO with ERM++ in Table 2 of the main paper. ERM++ substantially improves the performance of MIRO.

**DIWA Implementation:** We follow a simplified version of the DIWA (Rame et al., 2022) algorithm due to computational reasons; we average the parameters of the three seeds of ERM++, with shared initialization of the linear classifier. The authors of DIWA show that about half of the performance boost comes from the first few models averaged (Figure 4 of (Rame et al., 2022)), therefore this is a reasonable approximation of the method.

**CORAL Implementation:** We add the CORAL penalty term with a $\lambda$ value of 0.1, following Cha et al. (2022).

**Algorithm 1 ERM++:** Components of ERM++ are annotated in the algorithm comments. We run this training loop in two passes, the first to set training length by using a validation set split from the source domains. In the second pass we train on the combination of train and val data.

---

$ModelWeights \leftarrow AugmixWeights$                     ▷ Strong init
**while** $steps \neq LongTrainSteps$ **do**              ▷ Long Train
    $X, Y \leftarrow$ next(FullDataIterator)               ▷ Full Data
    **if** $steps \leq 500$ **then**
        Update linear classifier              ▷ Warmstart
    **else**
        Update linear classifier and backbone
    **end if**
    **if** $steps \geq 600$ **then**
        $ModelWeightsAvg \leftarrow$ Update          ▷ Weight Reg.
        $ModelWeightsAvgBN \leftarrow$ BN stats            ▷ UBN
    **end if**
**end while**

---

