# OpenReview forum: "ERM++: An Improved Baseline for Domain Generalization"
_ICLR.cc/2024/Conference — ICLR 2024 Conference Withdrawn Submission_

### Official Review · Reviewer_Xq2C · 2023-10-30

**Soundness:** 2 fair
**Presentation:** 3 good
**Contribution:** 2 fair
**Rating:** 3
**Confidence:** 4

**Summary:**

The paper proposes ERM++, which introduces several techniques to improve the ERM baseline on multi-source domain generalization. The techniques consist of that for training data utilization, model parameter selection, and weight-space regularization. Experiments on 5 widely-used domain generalization datasets with different backbones show the effectiveness of the method.

**Strengths:**

1. The paper investigates many techniques for the ERM model to improve multi-source domain generalization, which helps other researchers in this field to find suitable methods for their research.

2. The paper provides rich experiments to show the benefits of the utilized techniques for domain generalization.

**Weaknesses:**

1. Although the paper includes many techniques to improve the ERM model on domain generalization, most of these technologies have been proposed or are widely known. The improvement with these techniques is not surprising and inspiring.

2. The paper argues to propose a general baseline for future domain generalization works with the existing techniques. However the experiments, such as Table 3 (a), Table 4, and Table 5 show that different techniques benefit different datasets or settings, which is not general to the domain generalization task.

3. Except for the averaged accuracy, it is better to provide more insight of different techniques on how they benefit or harm the performance of domain generalization on different datasets.

**Questions:**

See weakness.

---

> ### Author Response · Authors · 2023-11-18
> **Author Response**
>
> > Although the paper includes many techniques to improve the ERM model on domain generalization, most of these technologies have been proposed or are widely known. The improvement with these techniques is not surprising and inspiring.
>
> While each technique may be well known, it is the combination of these techniques that constitutes ERM++. It is common for one technique to not compose well with another; for example CORAL does not compose well with our method(Table 3a.) Therefore, the combination of these techniques improving generalization performance is worth noting for the community.
>
> > The paper argues to propose a general baseline for future domain generalization works with the existing techniques. However the experiments, such as Table 3 (a), Table 4, and Table 5 show that different techniques benefit different datasets or settings, which is not general to the domain generalization task.
>
> > Except for the averaged accuracy, it is better to provide more insight of different techniques on how they benefit or harm the performance of domain generalization on different datasets.
>
>
>
>
> While each ERM component impacts different datasets in a different way, overall ERM++ improves over the ERM baseline on all datasets . See table 4; Row 7 is better than Row 1 in all cases, even though it doesn’t represent the best performance for each dataset individually.Domain generalization is a very challenging problem, and we cannot expect every technique to improve every dataset. Predicting which technique will perform well where in all DG situations in the future is an even more futile. Then, the best we can hope for is a set of simple techniques (simple also means less likely to overfit) that does very well on a number of known DG tasks, which is exactly what ERM++ accomplishes.

---

### Official Review · Reviewer_72G7 · 2023-10-31

**Soundness:** 2 fair
**Presentation:** 3 good
**Contribution:** 2 fair
**Rating:** 5
**Confidence:** 3

**Summary:**

A training protocol containing three main components is introduced to compose ERM++ which is a new baseline proposed in the submission. By observing the critical of the training length, a two-stage training procedure is conducted by determining the training length with the source domain validation set performance while training on the source training set in the first stage and then in the second stage training on the whole set. Model weight initialisation and weight-space regularisation, namely MPA application, are also studied. For the weight space regularisation idea, I hold my option for the later discussion. ERM++ is evaluated on a large variety of DG benchmarks with significant improvement comparied with vanilla ERM.

**Strengths:**

1. The experiments are dense and comprehensive. The proposed baseline is evaluated on most existing domain generalisation benchmarks comprehensively.
2. ERM++ explore existing practical technique tricks in DG with a detailed description.

**Weaknesses:**

1. The name is a little misleading as MPA is part of ERM++. If this is the case, it is natural to wonder what ERM + MPA performance will be like and by removing MPA from ERM++, then how ERM++ will perform. I think the closest setting is Table 4. Once MPA is added, for example comparing #1 and #2, the performance boosts significantly. The rest setting cumulates each tech one by one. But it is also important to know whether each one contributes independently.

2. One of the main points made in DomainBed is that without a complicated algorithm design with fair hyperparameter running ERM is a very strong baseline. However, ERM ++ is way more complicated than ERM.

3. Besides, since, it is justified by the submission that MPA works well with other introduced tricks, also it is good to know whether ERM++ is compatible with other advanced optimisation algorithms like SAM, GASM, SAGM, which is the benefit of using ERM.

4. In terms of the training cost, ERM ++ is compared with other models such as MIRO and DIVA, but the comparison with ERM is more important.

**Questions:**

See the above sections.

---

> ### Author Response · Authors · 2023-11-18
>
> > The name is a little misleading as MPA is part of ERM++. If this is the case, it is natural to wonder what ERM + MPA performance will be like and by removing MPA from ERM++, then how ERM++ will perform. I think the closest setting is Table 4. Once MPA is added, for example comparing #1 and #2, the performance boosts significantly. The rest setting cumulates each tech one by one. But it is also important to know whether each one contributes independently.
>
> Thank you for the suggestion of an ablative study in addition to a cumulative one. As each run is fairly expensive; we were unable to run an extensive ablation for the submission. It seems like the main concern is that some of the components may interact negatively with each other. We believe that a cumulative study suggests that this is not the case. Nevertheless, we will provide an ablative study for the camera ready and update the paper with it.
>
> > One of the main points made in DomainBed is that without a complicated algorithm design with fair hyperparameter running ERM is a very strong baseline. However, ERM ++ is way more complicated than ERM.
>
> Although we dissect ERM++ into many pieces, each piece is easy to implement. Then, we disagree with this assessment
>
> > I terms of the training cost, ERM ++ is compared with other models such as MIRO and DIVA, but the comparison with ERM is more important.
>
> ERM is also very expensive; the parameter turning tries 20 different trials of hyper-parameters. Instead, we use reasonable default ones for most values except training length.  This means that overall, ERM++ is 2.5x cheaper than ERM (we have two runs to select from training length (ES), each 4x longer(LT), so 20/8 = 2.5.

---

### Official Review · Reviewer_4v1r · 2023-10-31

**Soundness:** 2 fair
**Presentation:** 2 fair
**Contribution:** 2 fair
**Rating:** 5
**Confidence:** 3

**Summary:**

This work provides a detailed study of techniques used to improve the performance of empirical risk minimization (ERM) in domain generalization. Three categories of improvements are combined (data utilization, initialization, and regularization) to achieve state-of-the-art performance with lower computational cost than competing strategies.

**Strengths:**

1. The strategies used to maximize ERM performance for domain generalization are practically useful, and the resultant model is a competitive baseline for future work in the area.
2. Extensive experiments were performed with many different methods, architectures, and datasets. The ablation studies are also well done.
3. Careful analysis of the results were performed and edge cases were highlighted in the text. In particular, many of my initial questions were answered upon a closer read of the analysis, for example the discussion of VLCS performance in Section 5.1.

**Weaknesses:**

1. While the premise of the contribution - that ERM can match SOTA DG algorithms when appropriate data utilization, initialization, and regularization are applied - is important, the strong performance of ERM has been known since [1] and is not exactly novel. The main contribution of this paper is in applying recent “tricks of the trade” to further boost ERM numbers. While this may be helpful for practitioners, no new insight is offered as to why the proposed techniques are useful for ERM specifically or why ERM is a good baseline for domain generalization in the first place.
2. Many of the proposed improvements to ERM are not actually specific to ERM and can be applied to other state-of-the-art methods to improve performance. For example, it would be helpful to see a comparison where some methods in Table 2 are run with better initialization (say, AugMix) to see whether ERM still outperforms them. As is, the comparisons are made on somewhat unequal footing.
3. I believe the investigation of weight-space regularization is incomplete. First, why is ERM++ not run with SWAD, and is there any advantage of MPA in this scenario? Second, there is another category of weight-space regularization which is not included in the paper, namely sharpness aware minimization (SAM) [2] based techniques (e.g., SAGM [3]). SAM and SWA have been extensively compared [4] and found to each be beneficial in different circumstances. It would be interesting for the community to compare these two techniques in a DG setting, and I believe this experiment is necessary to claim a fully rigorous investigation of weight-space regularization for DG.

There is also a fair amount of confusing writing and typos, detailed in the next section.

**Questions:**

Here, I list some minor questions as well as suggestions for improving the writing.

1. The reference [5] cited in Table 2 should also be cited in the introduction.
2. The bar graph in Figure 1 is pixelated. If it was made with `matplotlib`, this can be fixed by setting the DPI or saving it as a PDF.
3. There is an inconsistent use of dataset names and abbreviations in the tables (e.g., TerraIncognita vs TI vs TerraInc vs TerraInco). I would recommend using the full name of the dataset everywhere, and perhaps reducing the font size when it doesn’t fit. The same goes for model names (e.g., Meal-V2 vs Meal V2 vs MV2).
4. There is an inconsistent use of spaces before citations, (e.g., Author(Citation) vs Author (Citation)). I encourage the authors to use the ~ LaTeX character to create a small space before the citations, and to keep this consistent throughout the text.
5. “Sketch” is misspelled in Table 6.
6. The headings in Table 7 are not explained. What are R0, R1, etc and P, I, Q, etc?

***Recommendation***

Overall, while this paper provides a useful benchmark on maximizing ERM performance for domain generalization, my concerns about novelty and the incomplete investigation of weight-space regularization cause me to lean slightly towards rejection rather than acceptance.

***References***

[1] Gulrajani and Lopez-Paz. In Search of Lost Domain Generalization. ICLR, 2021.

[2] Foret et al. Sharpness-Aware Minimization for Efficiently Improving Generalization. ICLR, 2021.

[3] Wang et al. Sharpness-Aware Gradient Matching for Domain Generalization. CVPR, 2023.

[4] Kaddour et al. When Do Flat Minima Optimizers Work? NeurIPS, 2022.

[5] Vapnik. An overview of statistical learning theory. IEEE Transactions on Neural Networks, 1999.

---

> ### Author Response · Authors · 2023-11-18
> **Author Response**
>
> >While this may be helpful for practitioners, no new insight is offered as to why the proposed techniques are useful for ERM specifically
>
> While each technique may be " a trick of the trade", it is the combination of these techniques that constitutes ERM++. It is common for one technique to not compose well with another; for example CORAL does not compose well with our method(Table 3a.) Therefore, the combination of these techniques improving generalization performance is worth noting for the community.
>
> > specifically or why ERM is a good baseline for domain generalization in the first place.
>
> Thank you for the very interesting question. While conclusive proof is beyond this paper, the authors have a main hypotheses.The main alternative to ERM is learning features invariant to domain label; which means that the domain cannot be predicted from the features of a sample. HOWEVER, classes are not balanced amongst domains in most of the common datasets; eg. domain 1 has a different class distribution than domain 2. This means that in order to become domain invariant, the network must learn to also suppress class-discriminative features, likely impeding performance.  Future investigation of this phenomenon is left to future work.
>
> > Many of the proposed improvements to ERM are not actually specific to ERM and can be applied to other state-of-the-art methods to improve performance. For example, it would be helpful to see a comparison where some methods in Table 2 are run with better initialization (say, AugMix) to see whether ERM still outperforms them. As is, the comparisons are made on somewhat unequal footing.
>
> A good point! We remind the reviewer that we propose a baseline, and we do not claim that ERM++ is the best possible method, just that it is a highly effective yet simple baseline. This means that it should be the job of researchers in the future to make sure any method they invent to compare to ERM++. Additionally, we direct the reviewer to table 3a, where we show that ERM++ (and hence the initialization with augmix) composes well with DIWA and MIRO.
>
> > I believe the investigation of weight-space regularization is incomplete. First, why is ERM++ not run with SWAD, and is there any advantage of MPA in this scenario? Second, there is another category of weight-space regularization which is not included in the paper, namely sharpness aware minimization (SAM) [2] based techniques (e.g., SAGM [3]). SAM and SWA have been extensively compared [4] and found to each be beneficial in different circumstances. It would be interesting for the community to compare these two techniques in a DG setting, and I believe this experiment is necessary to claim a fully rigorous investigation of weight-space regularization for DG.
>
> Again, thank you for the question. The authors of SWAD show that SAM and SWAD do not compose as well as As ERM and SWAD; in Table 4 of [1] they show that SAM + SWAD achieves 65.5% accuracy averaged across domains and datasets  while ERM+SWAD achieves 66.9%. Therefore, we leave it out of our comparisons. However, we agree that a more thorough investigations as to why would be very interesting, and will update our work with this for the camera ready.
>
> > There is also a fair amount of confusing writing and typos, detailed in the next section.
>
> Thank you for the suggestions, we will incorporate them into our writing.
>
> [1] Cha, Junbum, et al. "Swad: Domain generalization by seeking flat minima." Advances in Neural Information Processing Systems 34 (2021): 22405-22418. (https://proceedings.neurips.cc/paper_files/paper/2021/file/bcb41ccdc4363c6848a1d760f26c28a0-Paper.pdf)

---

### Official Review · Reviewer_Pkyt · 2023-11-01

**Soundness:** 2 fair
**Presentation:** 2 fair
**Contribution:** 2 fair
**Rating:** 3
**Confidence:** 5

**Summary:**

The paper present a new strong baseline, named ERM++, for the study of domain generalization (DG). By incorporating multiple previous results in training data utilization, parameter selection, and regularization, ERM++ achieved state-of-the-art performance on the DomainBed benchmark.

**Strengths:**

The obvious strength of the paper is the exciting results. The experimental settings are carefully described.

**Weaknesses:**

Despite presenting exciting results with elaborated experiments, the paper lacks technical insight into the effectiveness of various components, especially when they are used together. While I do see the merit of the engineering approach and agree that the field should appropriately acknowledge this as a baseline for large DomainBed, I do not think the current contribution of ERM++ is fit for a venue like ICLR. Thus, I cannot recommend acceptance for the paper.

**Questions:**

Why did the evaluation results for CMNIST and RMNIST not included in the paper?

---

> ### Author Response · Authors · 2023-11-18
> **Author responses**
>
> > While I do see the merit of the engineering approach and agree that the field should appropriately acknowledge this as a baseline for large DomainBed, I do not think the current contribution of ERM++ is fit for a venue like ICLR.
>
> Thank you for acknowledging the importance of engineering and  the importance of ERM++ to the field. We disagree, however, about the lack of technical insight. First, we discuss the motivation of each component in subsections of section 3. Furthermore, the  fact that we show these tricks work together in combination is itself a technical novelty, especially in as empirical a field as machine learning research.
>
> > Why did the evaluation results for CMNIST and RMNIST not included in the paper?
>
> MNIST are toy datasets compared to what we evaluate on; it is common to not include them in modern DG work, eg SWAD[1]
>
> [1] Cha, Junbum, et al. "Swad: Domain generalization by seeking flat minima." Advances in Neural Information Processing Systems 34 (2021): 22405-22418.